# Blood Pressure Measurement: From Cuff-Based to Contactless Monitoring

**DOI:** 10.3390/healthcare10102113

**Published:** 2022-10-21

**Authors:** Ping-Kwan Man, Kit-Leong Cheung, Nawapon Sangsiri, Wilfred Jin Shek, Kwan-Long Wong, Jing-Wei Chin, Tsz-Tai Chan, Richard Hau-Yue So

**Affiliations:** 1PanopticAI, Hong Kong Science and Technology Parks, New Territories, Hong Kong, China; 2Department of Computer Science and Engineering, The Chinese University of Hong Kong, Shatin, New Territories, Hong Kong, China; 3Department of Biomedical Sciences, King’s College London, London WC2R 2LS, UK; 4Department of Chemical and Biological Engineering, The Hong Kong University of Science and Technology, Clear Water Bay, Kowloon, Hong Kong, China

**Keywords:** blood pressure, hemodynamics, machine learning, deep learning, neural network, photoplethysmography, remote photoplethysmography

## Abstract

Blood pressure (BP) determines whether a person has hypertension and offers implications as to whether he or she could be affected by cardiovascular disease. Cuff-based sphygmomanometers have traditionally provided both accuracy and reliability, but they require bulky equipment and relevant skills to obtain precise measurements. BP measurement from photoplethysmography (PPG) signals has become a promising alternative for convenient and unobtrusive BP monitoring. Moreover, the recent developments in remote photoplethysmography (rPPG) algorithms have enabled new innovations for contactless BP measurement. This paper illustrates the evolution of BP measurement techniques from the biophysical theory, through the development of contact-based BP measurement from PPG signals, and to the modern innovations of contactless BP measurement from rPPG signals. We consolidate knowledge from a diverse background of academic research to highlight the importance of multi-feature analysis for improving measurement accuracy. We conclude with the ongoing challenges, opportunities, and possible future directions in this emerging field of research.

## 1. Introduction

Blood pressure (BP) is the measure of the force or pressure of blood pumped by the heart within the major arteries of the circulatory system, and it is influenced by multiple bodily factors such as cardiac output, vascular peripheral resistance, and arterial elasticity [1]. BP measurement usually accounts for two distinct values: systolic blood pressure (SBP) and diastolic blood pressure (DBP).

Elevated or high BP (hypertension) is known as “the silent killer” because it may show no symptoms until heart disease, failure, or damage occurs. Globally, high BP has accounted for approximately 12.8% of all global deaths per year and 3.7% of all disability-adjusted life years [2]. It is considered a major risk factor for cardiovascular disease (CVD), coronary heart disease (CHD), ischemic heart disease, atherosclerosis, myocardial infarction (MI), and hemorrhagic stroke. In Table 1, BP is categorized into several distinct stages identified by SBP and DBP levels, ranging from hypotension (low BP) to hypertensive crisis (high BP necessitating ICU treatment).

The importance of measuring BP as a gauge of cardiovascular health has long been recognized since the onset of allopathic medicine. The sphygmomanometer, invented in 1881, was the first blood pressure monitor that historically saw widespread usage [15]. Sphygmomanometers were reliable for the time but proved difficult for clinical practitioners to form a basic reading from the measurement. Generally, only systolic pressure could be measured from a mercury gauge, leaving the diastolic pressure to be inexactly estimated, and several redundant measurements needed to be taken for an accurate assessment [16]. Furthermore, the device was cumbersome and often caused mercury to spill on patients, leading to mercury poisoning and death in some cases [17].

Over the course of the 20th century, scientists would continue to develop and refine the sphygmomanometer, including how to read diastolic pressure and strengthen the frame of the device. However, in 1981, the first automated oscillometric manometer was developed, representing a major technological shift from manual hand pumps to digital algorithms [18,19]. This solved the accuracy problem that had plagued the older monitors of the mid-20th century and made them able to precisely measure the mean arterial pressure (MAP). However, over time, problems with automated oscillometric manometers emerged, including major discomfort toward children, the elderly, and the medically challenged [20]. The fitting of the cuff itself led to interference from either bodily vibrations or the individual’s motions to address the irritation posed by the cuff, and the device would lose accuracy over time [21]. In addition, it was able to measure the MAP but not the systolic or diastolic pressure, which needed to be calculated via subsequent mathematics. By the 1990s, countless jobs using the Internet allowed people to work from home, meaning patients would have to either transit to the nearest hospital or clinic for regular BP check-ups or operate an oscillometer at home, which many people at the time simply could not afford [22,23]. A convenient solution to these shortcomings was in high demand.

Photoplethysmography (PPG) has existed for almost 80 years. However, it was not until the 1980s that researchers truly began to understand its functions and potential for various medical applications (see [24,25,26,27,28] for reviews). PPG utilizes infrared (IR) light to measure the changes in volume of the vasculature. These volumetric changes can yield important information about cardiovascular status and health and can reliably estimate BP changes. PPG devices consist of a light source and sensor, where reflected light is measured as a response to changes in blood volume [29]. Most often, these are IR or light-emitting diode (LED) light sources, depending on the region of application.

Over the past four decades, PPG has been used to investigate hemodynamics, circulation patterns, and even changes in dental pulp [30]. Indeed, its application has often centered around the clinical investigation of atherosclerosis [31]. However, it was not until 1979 that the link between PPG and BP estimates was made in a study on venous insufficiency [32]. Thereafter, interest in PPG as an accurate and reliable measure of BP increased dramatically, especially toward the late 1980s, with many researchers using contact-based technology to increase the useful application of this method. However, as shown in Figure 1, interest in the field stagnated from the early 1990s to the 2000s, with far fewer related publications, possibly due to the technical limitations of the time.

A further challenge has been estimating BP using the pulse wave velocity (PWV), which is the speed needed for a pressure wave to travel a particular distance. To uncover the PWV, previous research attempted to measure the pulse arrival time (PAT), (The authors of [33] gave various definitions for PAT in their study. For instance, (1) ${PAT}_{\mathrm{Derivative}}$ is defined as the time difference between the R-peak of the ECG and the inflection point of the PPG, which is between the trough and the crest, and (2) ${PAT}_{\mathrm{Peak}}$ is defined as the time difference between the R-peak of the ECG and the crest of the PPG. ${PAT}_{\mathrm{Foot}}$ in [33] is equal to the PAT defined in this paper.) which is the time difference between the crest of the electrocardiography (ECG) waveform (called the R-peak) and the trough of the PPG waveform after time synchronization between the ECG and PPG waveforms [34]. The authors of [34] considered the mathematical relation between the PAT and BP. The authors of [35] proposed an algorithm based on the PAT for the continuous and cuffless estimation of the SBP, DBP, and mean arterial pressure (MAP) values.

However, the authors of [36] suggested that the pulse ejection period (PEP), defined as the electro-mechanical delay of the heart and the iso-volumic contraction time of the left ventricle. (Another equivalent understanding of the PEP is the time difference between the R-peak of the ECG and the opening of the aortic valve [33].) The authors of [37,38,39] suggested this should be subtracted from the PAT so as to give a more accurate estimation for the PWV. The authors of [40] also indicated that even though the PAT is easier to collect using wearable devices, using the PAT alone can result in errors due to the change in the PEP. The authors of [41] illustrated that it is more accurate to use PPG features to classify BP as normotension, prehypertension, or hypertension rather than using PAT features. This time difference, which can also be interpreted as the time taken for the pulse to travel along that artery, is called the pulse transit time (PTT), as shown in Figure 2, and it is given by
(1)PTT=PAT−PEP.
Hence, finding an accurate PEP temporarily became popular in the research field.

To solve the problem aroused by the PEP, the key is to find the timing of the aortic valve opening. There are investigations on the biological signals from various parts of a human body, as shown in Figure 3. The authors of [36] adopted impedance cardiography (ICG), which is a measurement using the first derivative of the thorax impedance change, to find the time difference between the point Q of ECG and the zero-crossing point of ICG as the PEP. Some studies also tried to directly find the PTT instead of the PEP. The authors of [42,43,44] measured the PTT instead of the PAT for BP estimation. The authors of [45] used ballistocardiography (BCG), which is a measurement of the reaction forces of the human body to cardiac ejection of blood into the aorta (instead of ECG) to bypass the calculation of the PEP and find the more accurate time difference for the PWV. The authors of [46] demonstrated that PPG waveforms for the foot and BCG can give better performance when tracking BP than the PAT. the authors of [47] tried to use seismocardiography (SCG), a noninvasive method developed in 1957 that records measurements of the thoracic vibrations of the heart [48] through an accelerometer on the chest to bypass the effect of the PEP, but there can be noises from variability in the morphology and artifacts from movement. Another method is called gyrocardiography (GCG), which uses the maximum point of the GCG waveforms in each cardiac cycle to identify the timing of the aortic valve’s opening. The authors of [49] adopted the GCG waveform from a gyroscope in a smartphone, SCG waveform from an accelerometer to extract the moments of aortic valve opening, and found the time difference between the PAT and PTT with the aid of a PPG waveform from an optical sensor with an audio function. The authors of [50] found the time difference between the point with a minimum gradient in the waveform and obtained an impedance phethysmography (IPG), which is a measurement of the amount of blood in blood vessels using an electric current passing through body tissues, and the point with the maximum gradient in a PPG waveform to calculate the PTT. The authors of [51] also adopted this method to find the PTT.

Although there are many viable measurement methods for BP, most of them require invasive devices or equipment that is inconvenient to carry. This motivates researchers to investigate whether it is possible to freely measure BP whenever a person wants. One feasible solution has been the use of remote photoplethysmography (rPPG). rPPG captures the subtle pixel changes of the light reflected from human skin by a digital camera such as a standard RGB camera and a webcam as shown in Figure 4. Since 2005, rPPG has being applied for various vital sign applications ranging from heart rate [53] to BP due to the outbreaks of SARS [54] and MERS [55]. Additionally, as highlighted in Figure 5, interest in remote BP measurement has steadily increased since 1996, which may have affected the rising curiosity in PPG. The necessity to continuously monitor an individual’s biological status, including BP, while remaining user-friendly was acknowledged more than ever as mass quarantines took effect globally [56,57].

More recently, researchers developed an iPhone application specifically for blood pressure monitoring to be used in tandem with the oscillometric finger press method to avoid the discomfort and potential life endangerment that come with automated cuffs [58]. Smartphones continue to establish themselves as the future of blood pressure monitoring technology, with online apps being a very convenient and portable sources of information not only for measuring the patient’s health but also to allow patients to communicate with clinical staff in order to adjust and personalize their healthcare. The authors of [59] reported that a research team developed and tested the prototype eyewear “Glabella”, containing optical sensors and able to operate on 24 h of battery life. The device reads the wearer’s pulse transit time (PTT) even during exercise or other common physical activities. Electronics company Samsung has developed their own smartphone application “S Health”, which is able to assess and monitor ${\mathrm{SpO}}_{2}$ levels using a built-in camera [60].

Because of the fact that there is a multitude of productive research progresses in BP measurement, it is useful if there are some reviews summarizing the key techniques and giving prospective future directions to motivate development. In particular, now that contactless BP measurement has emerged, it offers two options: an opportunity to (1) summarize what types of contactless BP measurement have been achieved and (2) examine possible implications for the industry to overcome future challenges. Thus, in this paper, we review the entire development of BP measurement and offer some insights for future research directions. We set up a flowchart in Figure 6 summarizing the research situation in BP measurement. This paper is organized as shown in Figure 7. Section 2 will outline the basic theory of the cardiovascular system, especially for how we can find BP. Section 3 will present the evaluation methods from PPG, including the non-machine learning (non-ML), traditional machine learning (TML), and deep learning (DL) methods, while Section 4 will present the counterparts from rPPG signals. Section 5 will discuss the strengths and weaknesses of the previous research, while Section 6 will discuss the perspectives of the research development of BP measurement. Finally, we draw our conclusions in Section 7.

## 2. Biophysical Theory

Investigation of BP has a very long history [61,62]. The first study on the wave nature of blood originated in [63], which derived the first formula for the pulse wave velocity (PWV). This concept was widely used afterward, as scientists tried to develop the relationship between the PWV and the properties of blood and the physiological structure. Moens [64,65] and Korteweg [66] derived a formula to express the PWV in terms of blood density ρ, arterial wall thickness *h*, the internal diameter *D*, and the Young’s modulus of the arterial wall *E* as
(2)$$PWV=\sqrt{\frac{Eh}{\rho D}},$$
which is called the Moens–Korteweg formula. (For the other mathematical derivation, see [67,68,69].) By assuming negligible elongation of the arteries [70], the PWV can be expressed by the distance of the artery from the heart to the periphery *l* divided by the pulse transit time (PTT), or the time taken for the pulse to travel along that artery (see [71] for a review on the PTT).

The authors of [72,73] derived a formula for the PWV in terms of the cross-sectional area of the artery *A*, the distending BP *P*, and the blood density ρ as follows:(3)$$PWV=\frac{l}{PTT}=\sqrt{\frac{A}{\rho }\frac{dP}{dA}}.$$
where $\frac{dA}{dP}$ in Equation (Equation 3) is the compliance of the arterial wall per unit length, (There is another term called the Windkessel compliance ($C_{W}=l_{\mathrm{eff}}\frac{dA}{dP}$), where $l_{\mathrm{eff}}$ is the effective length of the artery of the whole arterial system. This represents the sum of the compliance of the whole arterial system [74]. The value of $l_{\mathrm{eff}}$ depends on the height and weight of a patient [74], and the authors of [75] showed that $l_{\mathrm{eff}}=0.8$ m.) which is attributable to the material properties of the arterial wall. The relationship between *A* and *P* was initially examined in [74,76]. The formula is
(4)$$A=A_{\max}\left[\frac{1}{\pi }\tan^{-1}\left(\frac{P-P_{I}}{P_{II}}\right)+\frac{1}{2}\right],$$
which leads to the first derivative of *A* with respect to *P*, expressed as
(5)$$\frac{dA}{dP}=\frac{A_{\max}}{\pi P_{II}}{\left[1+{\left(\frac{P-P_{I}}{P_{II}}\right)}^{2}\right]}^{-1},$$
where $A_{\max}$ is the maximum cross-sectional area at high BP, $P_{I}$ is the position of the inflection point of BP, and $P_{II}$ is the width between points at the one-half and three-quarters amplitudes of BP, respectively. This is termed Wesseling’s model. By substituting Equation (Equation 5) into Equation (Equation 3) and taking into account the condition that the difference between *P* and $P_{I}$ is much larger than $P_{II}$ (i.e., $\left(P-P_{I}\right)>>P_{II}$) [71,77,78], we can obtain a simple relation between BP and PTT as follows:(6)$$BP=\frac{l}{PTT}\sqrt{\frac{2\rho P_{II}}{\pi +2}}+P_{I}.$$

This PTT–BP relationship does not consider a complicated physiology. For instance, smooth muscle contraction leads to a decrease in arterial compliance (per unit length) and thus the PTT (i.e., $P_{II}$ decreases, and thus the coefficient of 1/PTT decreases.). Collagen fibers gradually replace elastin fibers as one is aging, which causes a fall in the PTT (i.e., $P_{R}$ decreases as one’s age increases) [71,79]. However, the change in the PTT due to the aforementioned physiological reasons are negligible over a short time of measurement, which means that aging is only a factor when we consider the change in BP throughout the whole human life.

Apart from investigating the relation of BP∝1/PTT, most scientists propose models based on the Moens–Kortweg and Bramwell–Hill equations with an assumption-included function to relate BP and the PTT. For example, a popular model is
(7)$$BP=k_{1}\ln\left(PTT\right)+k_{2},$$
where $k_{1}$ and $k_{2}$ are some subject-specific constants. Another physical model, which can predict asymptotic behavior, is given by [80]
(8)$$BP=\frac{k_{1}^{\prime }}{{\left(PTT-k_{2}^{\prime }\right)}^{2}}+k_{3}^{\prime },$$
where $k_{1}^{\prime }$, $k_{2}^{\prime }$, and $k_{3}^{\prime }$ are some subject-specific constants. The experiments in [81,82] highlighted that BP is proportional to the inverse of the PTT instead of linearly related to the PTT over a wide range of BP. Even though some investigations adopted quadratic and nonlinear functions for the PTT [82,83,84] due to their great accuracy and reasonable asymptotic behavior, they require the evaluation of more than two unknown parameters, leading to more computation between BP and the PTT [71]. Hence, it is more convenient to use a linear model (Equation (Equation 5)) to find the BP.

Conventionally, the PTT is measured using the time delay between the proximal and distal arterial waveforms, (The authors of [59,85] reviewed different types of BP-measuring techniques and devices and those commercially available, with some related important historical developments. The authors of [86] described the principles of cuffless BP monitors and the current situation regarding BP monitor standards.) which can be found by calculating the time delay between the trough-to-trough (or foot-to-foot) parts of the PPG waveform from the two positions. This raised the popularity of investigating PPG to measure BP.

## 3. Contact-Based BP Measurement from PPG Signals

A traditional approach to finding BP is to analyze the shape of the PPG waveforms by extracting the features of the shapes (The authors of [87] conducted a comprehensive study of PPG-based authentication and discussed these applications’ limitations before pointing out future research directions. In addition, the authors of [71,88] reviewed the recent efforts in developing these next-generation blood pressure-monitoring devices and compared various mathematical models.) [24,25,26,27,28]. Previous studies were categorized into non-machine learning (non-ML), traditional machine learning (TML), and deep learning (DL) methods, as listed in Table 2. The corresponding results are listed in Table 2. The mean absolute error (MAE), root mean square error (RMSE), and mean error (ME) between the reference BP (${BP}_{r}$, also called the ground truth value or actual value) and predicted BP (${BP}_{p}$, also called the estimated value) were adopted to evaluate the experimental results. Unless specified, mathematically speaking, if there are *N* sets of references and predicted BP values, then they are defined as [89]
(9)$$MAE:=\frac{1}{N}\sum_{k=1}^{N}\left|{BP}_{r}\left(k\right)-{BP}_{p}\left(k\right)\right|,$$
(10)$$RMSE:=\sqrt{\frac{1}{N}\sum_{k=1}^{N}{\left[{BP}_{r}\left(k\right)-{BP}_{p}\left(k\right)\right]}^{2}},$$
(11)$$ME:=\frac{1}{N}\sum_{k=1}^{N}\left[{BP}_{r}\left(k\right)-{BP}_{p}\left(k\right)\right].$$
however, some scholars think that the error should be evaluated by subtracting the actual value from the estimated value. Based on this logic, each error should be found by ${BP}_{p}\left(k\right)-{BP}_{r}\left(k\right)$ for each integer *k*, which implies that the mean error should be defined as follows:(12)$$ME\;\left(\mathrm{Another}\;\mathrm{definition}\right):=\frac{1}{N}\sum_{k=1}^{N}\left[{BP}_{p}\left(k\right)-{BP}_{r}\left(k\right)\right]=-ME.$$

However, this only differs in that there is a minus sign between the two, and readers should pay attention to the definition of the error defined in the corresponding references.

In each entry in Table 2, we listed the datasets, their descriptions, the preprocessing methods, the model names, and their results. Only the preprocessing methods that affected the input features or model structure were mentioned, and the reduction of noise for the PPG waveform signals (Multi-various kinds of algorithms were adopted [90], including an LMS filter [91], moving average filters [92], adaptive filters [93,94], the Kalman filter [95,96], IMAR [97], wavelet analysis [98,99,100], median analysis [101,102,103], and a notch filter [104].) were excluded. Each method was tagged for three basic principles in BP prediction: (1) the PPG waveform, (2) PTT, and (3) personal biometric information. A method would be tagged if it considered the corresponding information.

**Table 2 healthcare-10-02113-t002:** Non-ML, TML, and DL models using PPG signals as inputs. The units for the SBP and DBP (unless specified) are mmHg. Acc. = accuracy, AE = absolute error, IQR = inter-quartile range, (1) = waveform features, (2) = PTT features, and (3) = personal information.

Year	Ref.	Dataset	Data Description	(1)	(2)	(3)	Preprocessing	Models	SBP	DBP
Non-machine learning (non-ML) methods										
2018	[105]	Private	32 subjects	•			Oscillometry		Error =3.3±8.8	Error =−5.6±7.7
2018	[58]	Private	18 subjects	•			Oscillometry		Error =−4.0±11.4	Error =−9.4±9.7
2016	[106]	Test: Private	85 subjects Smartphone data		•		-	AuraLife [107]	AE =12.4±10.5	AE =10.1±8.1
2018	[108]	Private	32 pregnant women Smartphone data				-	Preventicus	Error =5.0±14.5	-
2021	[109]	Private	965 subjects Smartphone data				-	Preventicus	Error =−0.41±16.52	-
Traditional machine learning (TML) methods										
2012	[110]	Private	5 subjects Smartphone data		•	•	-	Regression analysis	Acc. =97.45%	Acc. =97.63%
2013	[111]	Private	17 subjects Smartphone data	•		•	-	SVM	Acc. =100%	Acc. =99.29%
				•		•	-	Linear regression	Acc. =99.7%	Acc. =98.7%
2014	[112]	UQVS [113]	32 subjects	•		•	MIC feature selection	SVM	Acc. = 98.12%	Acc. = 97.22%
		Private	156 subjects Smartphone dataset	•		•			Acc. = 98.81%	Acc. = 98.21%
2016	[114]	Private	65 subjects (age =29±7, SBP =109.8±11.9, DBP =70.6±10.5)	•		•	Discrete wavelet transform, forward feature selection [115]	Nonlinear SVM	Error =4.9±4.9	Error =4.3±3.7
2016	[116]	UQVS [113]	32 subjects	•			-	SV Regression	AE =4.77±7.68	AE =3.67±5.69
2017	[117]	UQVS [113]	32 subjects	•			-	SVM	AE =11.64±8.22	AE =7.61±6.78
2017	[118]	Private	68 subjects (age =45±17, SBP =129±20, DBP =83±11)	•			-	Linear regression	MAPD =7.4%	MAPD =9.1%
2018	[119]	Private	7 subjects		•		-	Regression analysis	-	RMSE =5.2±2.0
2018	[120]	Private	205 subjects Smartphone data (independent splitting)	•			-	Lasso regression	AE =7.8±10.4	AE =6.1±7.1
2019	[121]	MIMIC	441 subjects	•			FFT, FFT${}^{-1}$,PCA	A series of 4 regressions	AE =3.97±8.901	AE =2.43±4.173
2020	[122]	[122]	15 subjects		•	•	-	Regression analysis	MAE =5.1	MAE =7.5
Deep learning (DL) methods										
2013	[123]	MIMIC [124]	-	•			-	ANN	AE =3.80±3.46	AE =2.21±2.09
2013	[125]	Train: MIMIC [124] Test: private (phone)	5 test subjects (SBP $\in \left[119,138\right]$, DBP ∈[63,75])	•			-	ANN	MAE =7.6	MAE =9
2015	[126]	MIMIC-II	4254 records	•			-	ANN	AE =13.78±17.46	AE =6.86±8.96
2016	[127]	Train: MIMIC-II Test: private	Train: 69 subjects Test: 23 subjects	•			-	ANN	Error =−1.67±2.46	Error =−1.29±1.71
2016	[128]	MIMIC-II	3000 subjects	•			-	ANN	AE =3.21±4.72	AE =4.47±6.85
2018	[129]	Private	84 subjects	•	•		-	LSTM	RMSE =3.73	RMSE =2.43
2018	[130]	Private	No subject description Exclude BMI >30	•			Activity features (this paper also considered the input features from the tri-axial accelerometer and not pure PPG methods)	LSTM	Median± IQR =5.95±2.05	Median± IQR =4.95±1.56
2018	[131]	MIMIC	120 subjects	•			Scalogram from CWT	GoogleNet	F1 score =82.95% for hypertension ${}^{3}$	
2019	[132]	MIMIC [124]	39 subjects	•	•			ANN-LSTM	0.93 r=0.9986	0.52 r=0.9975
2019	[133]	MIMIC-II	510 patients	•				ResNet [134] +GRU	MAE =15.41	MAE =12.38
			(independent splitting)	•		•			MAE =9.43	MAE =6.88
2019	[135]	MIMIC-II [126]	942 subjects (independent splitting)	•			-	CNN	AE =10.86±9.54	AE =5.96±5.60
				•	•				AE =9.30±8.85	AE =5.12±5.52
				•	•	•			AE =5.32±5.54	AE =3.38±3.82
2020	[136]	MIMIC-II	500 records	•			-	GRU	AE =3.25±4.76	AE =1.43±1.77
				•				LSTM	AE =3.23±4.74	AE =1.59±1.96
2020	[137]	Private	26 subjectsSmartphone data (various conditions)	•			-	CNN	AE =4.92±2.42	AE =5.28±1.80
2020	[138]	Figshare [139]	116 subjects(independent splitting, uniform subject distribution)	•		•		CNN	F1 score =40% for hypertension ${}^{3}$	
2020	[140]	UCI, MIMIC-II	1557 subjects	•	•		-	LRCN	AE =3.97±0.064	AE =2.30±0.196
2020	[141]	MIMIC-II [35]	942 subjects	•			-	PPG2ABP	MAE = 5.73	MAE = 3.45
2021	[142]	MIMIC-III	200,000 records (no description)	•	•		-	CNN-LSTM	AE =4.41±6.11	AE =2.91±4.23
2021	[143]	MIMIC-II	200 subjects (114 men, 86 women, age =61.6±14.6)	•	•		-	VGG19-LSTM	AE =1.73±4.96	AE =0.78±2.77
2021	[144]	Train [126] Test: UQVS [113]	5 train records (no description) 1 test subject (no description)	•		•	-	T2T-GAN	Error of AP =2.54±23.70	
2021	[145]	MIMIC-II UQVS [113]	20 subjects 32 subjects (subject independent splitting)	•			-	CNN-LSTM	AE =3.70±3.07 AE =3.91±4.78	AE =2.02±1.76 AE =1.99±2.45
2021	[146]	MIMIC [124]	48 subjects test split: 20% of total	•	•		-	CNN-LSTM	AE =1.2±1.6	AE =1.0±1.3
2021	[147]	MIMIC-III	Mixed: (12,000) records Non-mixed: (4000) records	•			-	AlexNet ${}^{2}$ [148]	MAE =8.8 MAE =16.6	MAE =4.9 MAE =8.7
2021	[147]	MIMIC-III	Mixed: (12,000) records Non-mixed: (4000) records ${}^{1}$	•			-	ResNet ${}^{2}$ [134]	MAE =7.7 MAE =16.4	MAE =4.4 MAE =8.5
2021	[147]	MIMIC-III	Mixed: (12,000) records Non-mixed: (4000) records ${}^{1}$	•			-	LSTM	MAE =11.6 MAE =16.4	MAE =6.7 MAE =8.6
2021	[149]	MIMIC I, III	100 subjects	•			-	U-Net	AE =3.68±4.42	AE =1.97±2.92
2021	[150]	MIMIC II [35]	942 subjects	•			-	U-Net	MAE =5.16	MAE =2.89
2021	[151]	(dataset)	(subjects)				-	LASSO-LSTM	MAE =4.95	MAE =3.15
2021	[152]	MIMIC-II [126]	5289 subjects	•			-	LSTM Autoencoder	AE =4.05±5.25	AE =2.41±3.17
2022	[89]	Train: MIMIC-II Test: MIMIC-II Test: UQVS [113]	MIMIC-II: (12,000) records Test: random, 3000 records UQVS: 32 subjects	•	•		MFMC filter	MLPlstm-BP	AE =3.52±5.09 AE =4.39±6.43	AE =2.13±3.07 AE =2.54±3.76
2022	[89]	train:MIMIC-II test: MIMIC-II test: UQVS [113]	MIMIC-II: (12000) records Test: random 3000 records UQVS: 32 subjects	•	•		MFMC filter	gMLP-BP	AE =4.18±5.87 AE =4.73±7.11	AE =2.47±3.52 AE =2.69±4.10
2022	[153]	[147] Non-mixed [126]	[147]: 1,250,000 samples ${}^{1}$ [147]: subject uniform distribution [126]: 4254 records	•			-	InfoGAN ${}^{2}$ [154] enc-dec	MAE =14.26 AE =10.59±9.07	MAE =7.11 AE =5.95±5.76

^1^ For the non-mixed dataset, the train and test splitting is subject independent, where each subject contributed no more than 2000 samples, and there were approximately 150 million
samples in total. ^2^ Modified architecture. ^3^ Result of non-HT vs. HT classification.

### 3.1. Non-Machine Learning (Non-ML) Methods

#### 3.1.1. Mathematical Modeling

To date, there are some areas of research that have specifically focused on mathematically studying BP. The first trial involved in the study of BP and blood flow was the Windkessel model (This model is also called the lumped parameter model. See [155,156,157,158,159,160,161,162] for extensions of the model and [163] for a review.) [164], which adopted the concept of electric currents to stimulate the cardiovascular system (CVS) of the human body. The circuits considered in [164] contained a constant resistor representing the resistance of the peripheral artery and a capacitor representing the compliance of aortas. The authors of [165] extended the model in [164] by adding an extra resistor to model the characteristic impedance of the aortas in order to obtain the behavior of the systemic input impedance from medium to high frequencies. The authors of [166,167] highlighted that the four-element Windkessel model with an inertial term in parallel with the characteristic impedance is better than the three-element counterpart (There are two versions of the three-element counterpart raised in [168] and [169], respectively.) for describing the behavior of the entire systemic tree and modeling the vascular properties via estimating the suitable parameters.

However, several Windkessel models failed to explain the phenomena of pulse wave propagation and reflection throughout the arterial system, which meant that new models with pulse wave propagation were necessary. The first contribution toward mathematical modeling with pulse wave propagation was from Womersley [170], who first investigated the PWV profile in a rigid tube filled with a viscous fluid. The authors of [171] examined the PWV profile through an elastic tube with a finite viscosity and developed a mathematical analysis of blood flow in the arteries. The authors of [172] tested various materials that are possible to use for mimicking the textures of blood vessels and arteries, as well as the effects of stress and temperature on the textures. The authors of [173] found that elastic tubes with a nonlinear stress/strain ratio obtained the most similar performance to that of real arteries and blood vessels.

After investigating the properties of blood vessels, researchers endeavored to model the blood vessel system with fluid mechanics equations. The authors of [174] modeled the small arteries and arterioles as a structured tree to derive the mathematical expression of the root impedance of the structured tree in the frequency domain based on a linearixed Navier–Stokes equation. The study of blood flow and BP was carried out by using a nonlinear one-dimensional (1D) model [175,176,177,178], a three-dimensional (3D) model [179,180], and other mathematical and physical models [174,181,182,183]. The authors of [68] proposed a mathematical model for BP for different age groups. The authors of [184] developed a biophysical model of CVS simulating blood flow in the upright position of the body in order to investigate the effect of gravity on the PWV. The authors of [69] investigated the propagation of pulses through an elastic tube filled with viscous fluid under initial pressure using Navier–Stokes equations. In addition, the authors of [185] proposed pulse decomposition analysis using hyperbolic secant (sech) waves and suggested that hyperbolic secant wave decomposition gives more accurate blood pressure values than the Gaussian function.

#### 3.1.2. Direct Verification Methods

Apart from the mathematical models, there have been further studies directly examining the feasibility of BP measurement with the use of some specific mobile apps and smartphones. In these cases, researchers were unaware of the algorithms due to a range of confidential reasons. The authors of [186] utilized two contact-based cameras for simultaneous acquisition of PPG from the fingertip of the index finger and the forehead temple, and they revealed that the correlation between the PTT and OFP (OFP is the time interval between the minimum PPG from the temple and the maximum PPG from the fingertip.) was 0.86±0.06. The authors of [187] presented algorithms that can be executed directly on current smartphones to obtain clean and robust heart sound signals and to extract the pulse wave characteristics. The authors of [106] examined the instant blood pressure (IBP) estimation app AuraLife [107] and pointed out that its sensitivity for hypertensive BPs was only 0.22. The authors of [105] indicated the results of BP measurement by pressing one’s fingers against a smartphone. The errors for the SBP and DBP were 3.3±8.8 mmHg and −5.6±7.7 mmHg, respectively. The authors of [58] performed the same verification with an iPhone instead of a smartphone. The errors for the SBP and DBP were −4.0±11.4 mmHg and −9.4±9.7 mmHg, respectively. The authors of [119] developed and evaluated a smartphone-based BP-monitoring application called Seismo, which measures the time between the opening of the aortic valve and the pulse reaching the peripheral artery. The RMSE of the DBP was 5.2±2.0 mmHg. The authors of [122] estimated BP using the PTT calculated from PPG and a phonocardiogram (PCG) recorded using a microphone. (This experiment originally used the rear camera of a smartphone to record the HR and ${\mathrm{SpO}}_{2}$ and a microphone to record PPG and the phonocardiogram (PCG).) The authors of [108,109] studied the app Preventicus for BP measurement.

Thus, we note that mathematical methods involve solving various differential equations and assuming the material properties of the arteries, while direct verification methods check the feasibility of the algorithms in BP measurement without knowing the algorithms. Next, we will discuss the data-driven methods, and given the data we obtained before, we are going to find what the model will be. Initially, we will examine the TML methods of PPG.

### 3.2. Traditional Machine Learning (TML) Methods

Previous research on BP measurement in TML is mainly feature-based, as shown in Figure 8. Those algorithms extract specific features of the signal (PPG) based on physiological motivations or nontrivial statistical metrics and combine them as linear or nonlinear regression and SVM models to predict BP. The authors of [188] examined the relationships between arterial blood pressure (ABP) and certain features of PPG signals obtained from 15 young healthy subjects under 3 experimental phases: rest, a step-climbing exercise, and recovery from the exercise. They investigated four features of PPG, namely the width of a $\frac{2}{3}$ pulse’s amplitude, width of a $\frac{1}{2}$ pulse’s amplitude, systolic upstroke time (SUT), and diastolic time (DT), to find the feature that possessed the highest mean correlation coefficient with BP and then set up a simple linear regression connecting that feature and BP to estimate the BP. It was found that the DT had the highest correlation with BP. The mean differences for SBP were 0.21±7.32 mmHg, and those of DBP were 0.02±4.39 mmHg. The authors of [189] presumed that there are some cardiovascular peculiarities in old age. To verify this point, they first separated their data into two groups: a group with patients aged 60 and over and a group with patients younger than 60 years. Then, they estimated the SBP of different groups by multiple regression analysis from the information and features of the PPG signals of each subject. It was shown that the SD of the classified data can be 5.5 mmHg better than that of the non-classified data. The authors of [110] introduced two methods to record the PTT. (This paper uses the vascular time interval (VTT), which is the time difference between the first heart sound of a phonocardiogram (PCG) and the systolic peak in PPG after time synchronization of all considered vital signals.) A regression analysis was built to estimate the SBP and DBP with parametrization by personalized factors, such as weight and height. The authors of [111] adopted 14 time domain features of the PPG waveform in addition to those personalized factors, such as height, weight, and age, as inputs to train a model which mixed linear regression and SVM for classifying BP at different levels.

Later, researchers started to find useful features based on statistical metrics and signal processing methods instead of the shape features of PPG. In order to reduce the dimensionality of input data and cut down the resource requirements, the authors of [112] adopted the maximal information coefficient (MIC) to select 12 time domain features and 7 frequency domain features of the PPG waveform and trained an SVM to classify the BP values into various bins. The authors of [190] proposed an SVM to estimate continuous BP from the heart sound signals obtained by the microphone of a smartphone. The absolute errors (AEs) of the SBP and DBP were 4.339±6.121 mmHg and 3.171±4.471 mmHg, respectively. The authors of [114] extracted the frequency features by a discrete wavelet transform (DWT) and fed them into a linear SVM to predict the BP. The absolute error (AE) of the DBP was 4.6±4.3 mmHg, while that of the SBP was 5.1±4.3 mmHg. The authors of [116] applied several analytical techniques, including random error elimination, adaptive outlier removal, MIC, and Pearson’s correlation coefficient-based feature assessment. The optimum results were 4.77±7.68, 3.67±5.69, and 3.85±5.87 mmHg for SBP, DBP, and MBP, respectively. The authors of [191] validated pulse wave analysis (PWA) based on a multi-parameter model by using MIMIC physiological data and comparing the corresponding results with the pulse arrival time (PAT) method. Those experimental data of 23 subjects over a day were sent to a regression model for BP prediction, which gave 10.6±3.3 mmHg (PAT) and 8.7±3.2 mmHg (PWA) for SBP and 6.0±2.3 mmHg (PAT) and 4.4±1.6 mmHg (PWA) for DBP, respectively.

Researchers also proceeded to increase the number of features as much as possible to fine-tune the accuracy. The authors of [117] extracted more than 7000 heartbeats and 9 parameters as the input vector for training the SVM. It demonstrated better accuracy than the linear regression method and better accuracy than the ANN method for DBP. The authors of [118] used five features in total—the ratio of PPG features, SUT, the inverse of SUT squared, age, and body mass index—to train a gradient descent learning-based linear regression with a series of post-processing methods to reduce the unsuitable data, leading to minimum absolute percentage differences (MAPDs) of 7.4% (SBP) and 9.1% (DBP), respectively. The authors of [120] integrated 233 features in the time and frequency domains to train an ensemble of models based on demographic and physiological partitioning. The AE of DBP was 5 mmHg, and that of SBP was 6.9 mmHg. The authors of [192] developed a far simpler cuffless method using only the heart rate (HR) and modified normalized pulse volume (mNPV), which can be measured using a smartphone, based on the fact that BP is equal to the product of the cardiac output (CO) and total peripheral resistance (TPR), where the CO and TPR are correlated with the HR and mNPV, respectively. The SBP and DBP are estimated by nonlinear regression. The errors of the SBP and DBP were 0.67±12.7 mmHg and 0.45±8.6 mmHg, respectively.

The authors of [121] estimated BP based on a series of four regressions (decision tree regression, support vector regression, adaptive boosting regression, and random forest regression) of whole-based feature extraction, leading to the AEs of SBP and DBP being 2.43±4.173 and 3.97±8.901 mmHg, respectively, on the MIMIC-II dataset. The authors of [193] used the PTT to estimate one’s SBP and DBP by linear regression after using filtering and peak detection algorithms in order to reduce the noise of PPG. The AEs of the SBP and DBP were 2.07±2.06 mmHg and 2.12±1.85 mmHg, respectively, which are lower than the BP estimation standard (5±8 mmHg). The authors of [194] adopted the time, frequency, and the time–frequency domain features of PPG signals from 219 subjects, undergoing preprocessing and feature extraction steps to feed the regression models to predict the SBP and DBP. The RMSEs of SBP and DBP were 6.74 mmHg and 3.59 mmHg, respectively.

Up to this point, we can see that TML methods have mainly been feature-based. Those features closely related to the blood pumping mechanism or having high correlations with BP are frequently selected as inputs. The models are mainly linear or nonlinear regression and an SVM. Regression is mostly used for finding the values of BP, while an SVM is mostly used for finding the categories of BP. Next, we will review how BP is measured based on deep neural network (DNN) methods.

### 3.3. Deep Learning (DL) Methods

There are many suggestions for DNN models, and the corresponding results satisfy the standards of the British Hypertension Society (BHS) [195], Association for the Advancement of Medical Instrumentation (AAMI) [196], and Institute of Electrical and Electronics Engineers (IEEE) for blood pressure measurement devices. The BHS standard assigns a grade by the percentage of test samples which have an absolute error less than 5 mmHg, 10 mmHg, and 15 mmHg. The criteria are presented in the Table 3. The AAMI standard is passed when the mean error (ME) and SD of the error are within ±5 mmHg and ±8 mmHg, respectively.

#### 3.3.1. PPG Waveform-Based Methods

Passing the human-defined features to a basic, fully connected neural network (PCNN) (This network is also known as a fully connected neural network (FCNN) and multilayer perceptrons (MLPs).) was a common early approach to applying neural networks to BP predictions. The input features usually describe the waveform patterns of PPG signals. The output layer usually contains two real-valued outputs: the predicted SBP and the DBP. This approach is similar to the regression method, which also utilizes human-defined features to perform regression analysis. The upper part of Figure 9 represented the structure of these methods. Better performance is expected when replacing the regression model with an ANN. The authors of [197] compared the results for BP predictions between an ANN and different regression methods, and the ANN reported the best performance.

The authors of [123] defined 21 features of the PPG waveform and passed them through an ANN. The features included the systolic and diastolic width at every 25% and 33% of the pulse height, respectively, as well as the cardiac period (CP), SUT, and DT. The AAMI standard was met with mean errors of 3.80±3.46 and 2.21±2.09 mmHg for SBP and DBP, respectively. As an application of this work, the authors of [125] utilized an ANN to deal with smartphone PPG with the aid of a preprocessing algorithm. However, their results were above the AAMI standard. Other researchers also followed this approach to utilize ANNs with different feature definitions. The authors of [127] used a fast Fourier transform to extract the frequency domain features, which should be more stable than the time domain features since they do not require alignments. The properties in the derivatives of the PPG waveform and differences between the PPG cycles were considered [128].

The feature extraction strategies for ANN models usually consider a cycle of the PPG signal or average several cycles to represent the signal. The authors of [128] mentioned the importance of the difference between cycles, which the ANN model may overlook. Recurrent neural networks (RNNs), including GRU and LSTM, are capable of taking into account the difference between cycles as well as the time domain variation of the input features. The authors of [136] compared the results between multiple linear regression, an ANN, GRU, and LSTM for BP estimation from seven selected PPG waveform features and indicated that GRU and LSTM adequately outperformed the non-recurrent models.

The predefined feature may not capture all the details in the signal. With the progression of deep learning, DNNs give outstanding performance when acting as a feature extractor. The authors of [133] directly passed the raw PPG signal and its first and second derivatives to a model combined with the ResNet block and GRU. Convolutional neural networks (CNNs) such as ResNet usually act as a feature extractor to automatically obtain the waveform features from the raw signal input. The lower part of Figure 9 represents the structure of the models that support end-to-end BP estimation from raw PPG signals. Many recent works use a CNN or the popular CNN-RNN architecture to estimate BP from raw PPG signals. Baek et al. [135] proposed a model utilizing dilated and strided convolution to extract the time and frequency features of PPG. They also applied the model to smartphone PPG data [137], and the result passed the AAMI standard and was comparable to the method using the PPG sensor dataset. The authors of [145] proposed a CNN-LSTM model for PPG-based BP estimation. Their results obtained a Grade A rating on the BHS standard and passed the AAMI standard for SBP and DBP.

Because one of the primary goals of BP prediction is to identify hypertension (HT) patients, some research works evaluated their methods in classification approaches and focused on hypertension sensitivity. The authors of [131] entered a 2D scalogram into GoogLeNet and performed BP classification using the continuous wavelet transform (CWT), as shown in Figure 10. The authors of [138] proposed a novel CNN model which performs BP estimation and classification simultaneously. The estimated BP would contribute to correcting the final predicted BP class. They reported a 90% F1 score on the non-HT versus HT binary classification task. The F1 score for NT, PHT, and HT classification was 40%, which was already the best result among the related works. The model also considers personal biometric features, which we will focus on in Section 3.3.3.

Some recent research has focused on the specific area of continuous BP waveform prediction. This task aims to transform the PPG waveform into an ABP waveform. In 2020, the authors of [141] proposed PPG2ABP, the first algorithm to predict continuous BP. Their method is composed of two networks: a U-Net-based approximation network to transform the PPG to ABP and a MultiResNet model for refinement (Figure 11). PPG2ABP obtained a grade of A for DBP and a grade of B for SBP according to the BHS standard. The authors of [149] proposed a model that can be implemented on wearable devices in the extended works that used the U-Net structure for signal translation, and the authors of [150] used self-supervised learning to reduce the training computation cost. Apart from the CNN-based architecture, researchers also adopted generative models for the signal translation task. Inspired by CycleGAN, the authors of [144] proposed the T2T-GAN model. Their work is capable of bidirectional signal translation between PPG and ABP signals. The authors of [152] proposed an LSTM-based autoencoder model which replaces the ANN in the original autoencoder model with LSTM. The encoder was pretrained and unsupervised, as with the original autoencoder, and then the decoder was trained for ABP prediction. Their model obtained a grade of A according to the BHS standard and passed the AAMI standard for both SBP and DBP, making it the most accurate model for continuous BP predictions currently.

#### 3.3.2. PTT-Based Method

As the PTT has been proven to be very effective for increasing the accuracy of traditional BP prediction methods, it is intuitive to construct a deep learning model that can consider the PTT. Therefore, many items of research input both PPG and ECG signals to models for BP prediction. This approach can not only extract the waveform features for both signals but also capture the time difference between the two signals, which implies the PTT features. The authors of [129] selected seven predefined features from ECG and PPG, including the PTT as measured from two signals, and passed them into a deep bidirectional LSTM network. Unlike traditional PTT-based methods, their model can get rid of the long calibration period. Their best RMSEs for SBP and DBP were 3.73 and 2.43 mmHg, respectively, outperforming the only PPG-based method in the same period.

With the rise of CNNs and feature learning, the CNN-RNN structure was also widely adopted in PTT-based methods. Figure 12 represents the structure of PTT-based methods that can automatically extract both waveform features and PTT information from ECG and PPG signals. The authors of [132] proposed an ANN-LSTM network with PPG and ECG as the inputs, where an ANN acts as the feature extractor. The mean absolute error (MAE) was 1.10 and 0.58 for SBP and DBP, respectively, largely outperforming all the PPG-based methods. Later, there were different variants or advanced models in the CNN-LSTM architecture [140,142,143,146]. Jeong et al. [146] reported AEs of 1.2±1.6 and 1.0±1.3 mmHg for SBP and DBP, respectively, which are extremely close to the ground truth.

There are also some models other than that in the CNN-RNN architecture for PTT-based BP prediction. Inspired by the MLP-Mixer in computer vision, the authors of [89] proposed MLP-BP. The ECG and PPG signals are preprocessed by a novel multi-filter-to-multi-channel (MFMC) algorithm, which stacks 12 differently filtered signals as the input. They also implemented different variants of MLP-BP. Two of the variants containing the LSTM layer, MLlstm-BP and gMLP-BP, reported better accuracy.

#### 3.3.3. Personalization Factors in Deep Learning Methods

For the TML methods of BP prediction, it was common to consider personal biometric features in the regression model or predefined features, such as age and BMI. Among the TML methods we have introduced, the authors of [111,112,114] gave the waveform-based methods that include biometric features, and the authors of [110,122] presented the PTT-based methods that include biometric features. For the DL methods, the models always focus on extracting the features from the signal waveform, and fewer researchers would take into account the personal information. As one of the few DL methods that consider personal factors, the authors of [138] packed the BMI information with the embedding after the convolutional layers and before the prediction layers (MLP) as shown in Figure 13. This method shares an idea with TML methods, since BMI is regarded as one of the features for the predictor. Figure 14 shows the whole structure of their model, including the integrated class assessment algorithm. Recently, fine-tuning the model using subject-specific data has been used as a personalization technique in some DL methods [133,135,147], which we will introduce in Section 6.

## 4. Contactless BP Measurement from rPPG Signals

Remote photoplethysmography (rPPG) is a technique that adopts the specular and diffused components of incident light to measure physiological signals, as shown in Figure 4. Since the diffused component of incident light carries information on subtle blood flow volume changes under human skin, it is possible to analyze that component to find the small light variations in blood flow so as to obtain live data of vital signs such as the heart rate [53], respiration rate [200], blood pressure [201,202], and oxygen saturation [200]. However, remote BP prediction also has susceptibilities in handling noise and artifacts unrelated to the hemoglobin signals, such as makeup [203], skin tone [204,205], illumination [206,207], camera distance [208,209], camera specification [210,211], and subject motion [212,213]. Considering the pros and cons of using rPPG signals, several remote techniques have been developed to manipulate deficiencies using a sophisticated solution to estiTNLmate BP from the rPPG signal. Previous studies are categorised in non-machine learning (non-ML), traditional machine learning (TML) and deep learning (DL) methods, as listed in Table 4, and the corresponding results are also listed in this table.

**Table 4 healthcare-10-02113-t004:** Non-ML, TML, and DL models using rPPG signals as inputs. The units of SBP and DBP (unless specified) are mmHg. AER = average error rate, AE = aboslute error, (1) = waveform features, (2) = PTT features, and (3) = personal information.

Year	Ref.	Dataset	Data Description	(1)	(2)	(3)	Preprocessing	Models	SBP	DBP
Non-machine learning (non-ML) methods										
2015	[214]	Private	10 subjects		•		-	-	$\bar{r}=-0.879$	-
2016	[215]	Private	7 subjects		•		iPTT	correlation estimation	$\bar{r}=-0.80\pm 0.12$	-
2019	[216]	Private	20 subjects		•		-	Time difference between 2 waveforms from a palm	$\bar{r}\approx 0.6$	-
2020	[217]	Private	6 subjects		•		Modeling rPPG by Gaussian curves, pair filtering	Regression	AE =8.32±8.81	AE =12.34±7.10
Traditional machine learning (TML) methods										
2016	[218]	Private	45 subjects	•			PCA [219]	Regression	AE =3.90±5.37	AE =3.72±5.08
2016	[220]	Private	3 subjects		•		ICA	Linear regression	AE =9.48±7.13	AE =4.48±3.29
2017	[221]	Private	13 subjects (SBP =110.03±11.05, DBP =72.78±7.50		•		iPTT	KNN model with transfer learning	RMSE =14.02	RMSE =7.38
2017	[222]	Private	45 subjects		•		PWV formula	2nd order polynomial regression	AE =4.22±5.15	AE =3.24±2.21
2018	[223]	Private	8 subjects having individual models	•		•	ICA	Linear regression	MAE of MBP $\in \left[1.50,4.15\right]$	
2019	[224]	Private	10 subjects	•			Pulse wave detection	Lasso regression	Error of BP =−1.0±5.6	
2019	[225]	Private	100 subjects (70 men and 30 women, age $\in \left[22,50\right]$)	•		•	JADE algorithm [226,227]	multiple linear regression	RMSE =4.1665	RMSE =2.8531
2021	[228]	Private	191 subjects (141 men and 50 women, age $\in \left[20,61\right]$)	•			Green channel, cheek and nose areas, Mallat algorithm, peak extraction	Support vector regression	AE =9.97±3.35	AE =7.59±2.58
Deep learning (DL) methods										
2017	[229]	Private	20 subjects without known blood pressure disease	•			ICA	Feedforward neural network	AER (afternoon) =9.62% AER (evening) =8.4%	AER (afternoon) =11.63% AER (evening) =11.18%
2019	[230]	Private	1328 subjects (SBP $\in \left[100,139\right]$, DBP $\in \left[60,89\right]$)	•	•	•	TOI, PCA	ANN	Error =0.39±7.30	Error =−0.20±6.00
2021	[147]	Private	50 subjects, subject independent splitting	•		•	Pretrained by PPG	AlexNet ${}^{1}$ [148]	MAE =14.2	MAE =10.7
2021	[147]	Private	50 subjects, subject independent splitting	•		•	Pretrained by PPG	ResNet ${}^{1}$ [134]	MAE =12.7	MAE =10.8
2021	[147]	Private	50 subjects, subject independent splitting	•		•	Pretrained by PPG	LSTM	MAE =14.4	MAE =10.5
2022	[153]	Private	Train: 961 subjects Test: 177 subjects	•		•	CHROM [231]	InfoGAN ${}^{1}$ [154] Encoder-decoder	AE =9.13±8.18	AE =8.76±6.13
2022	[232]	Private	10 subjects, subject mixed splitting	•			2 spatial descriptors	ResNet [134] +CBAM [233] ${}^{1}$	MAE =6.7 r=0.81	MAE =5.4 r=0.84

^1^ Modified architecture.

**Figure 14 healthcare-10-02113-f014:**
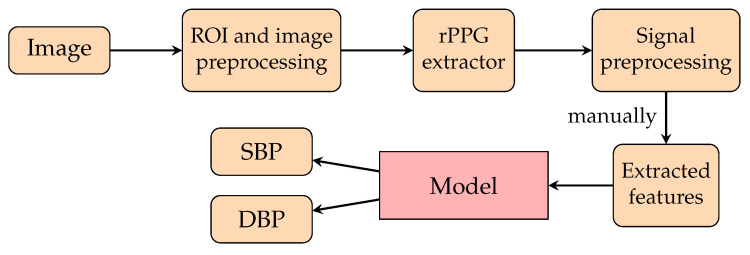
A pipeline of a deep learning model for rPPG singals proposed in [147,229,230]. The pixels of the ROIs of face images are preprocessed and then extracted to form rPPG signals. These rPPG signals are then preprocessed, and those features of preprocessed rPPG signals are extracted to feed the respective models for training to predict SBP and DBP signals.

### 4.1. Non-Machine Learning (Non-ML) Methods

To take advantage of the remote methods, multiple areas of the body are used to predict BP. The authors of [215] revealed that SBP has a strong correlation with the PTT through sequential images of faces and hands obtained by a high-speed camera (420 fps). Another study that utilized the PTT to estimate BP is [214]. The authors of [234] proposed a non-contact technique to estimate the BP variability using a pulse wave obtained via imaging photoplethysmography (iPPG) to find the phase differences (PDs) between two iPPG waveforms. The cross-correlation between SBP and PD was calculated, and it was demonstrated that the PD had a higher correlation with SP compared with a time difference method. However, the phase difference was distorted by the skin’s surface and may not be a truly accurate prediction for BP.

A disadvantage of the PTT is the restriction to maintain the relative position of the two skin regions during video recording. To overcome this disadvantage, the authors of [216] aimed to estimate BP from a region of interest (ROI) in a skin area. They calculated the new index ($t_{BH}$), which is a time difference between a corner point of the raw waveform from videos of the palm and a minimum point of a band-pass-filtered signal’s minimum values in a band-pass filtered waveform from the same video. Here, $t_{BH}$ has similar accuracy to the PTT approach; however, $t_{BH}$ can overcome the video restriction in the PTT approach.

Since face motion artifacts can cause the poor quality of rPPG, the authors of [217] developed pair filtering to choose two rPPG waveforms from different regions of the face and hand to estimate the PTT. After choosing the suitable rPPG, the authors of [217] proposed a model to estimate PPG from rPPG by using two Gaussian curves. This model used for calculating the PTT. This study showed that the proposed model with a filter pair can improve the correlation between the PTT and BP. It used a machine learning model to stimulate PPG instead of estimate the BP.

### 4.2. Traditional Machine Learning (TML) Methods

To continue from Section 4.1, applying traditional machine learning techniques to calculating BP is one of the interesting topics in the research field. To take advantage of the correlation between PTT and SBP, there are some investigations that apply the principle of PTT with a lightweight machine learning model to calculate BP. The authors of [221] proposed a KNN model to calculate BP by using the PTT from the rPPG waveform of the face and palm. The model was trained by transfer learning with the MIMIC II dataset first, followed by training with a private rPPG dataset. The RMSEs of SBP and DBP were 14.02 mmHg and 7.38 mmHg, respectively. Since it was illustrated in [222] that the PWV is strongly correlated with BP by the Moens–Korteweg equation, second-order polynomial regression using the PWV as the input layer was applied. Instead of inputting the PTT directly into the model, the PTT is measured to calculate the PWV, and the PWV is taken into account for calculating BP. The AEs of SBP and DBP were 4.22±5.15 mmHg and 3.24±2.21 mmHg, respectively.

Since the optical absorption of the human skin depends on the hemoglobin, melanin, and shading components, the rPPG waveform is affected by noise. To make rPPG more effective, it is better to separate the noise and the rPPG signal. The authors of [235] introduced a cleaning method as a linear combination of all three color channels of RPG and defined three independent signals with independent component analysis (ICA). The authors of [220] proposed a linear regression for predicting BP, as shown in Figure 15. It used ICA to clean the noise and applied PTT techniques as the input layer of the regression model. It obtained AEs of 9.48±7.13 mmHg for SBP and 4.48±3.29 mmHg for DBP. The authors of [223] also applied ICA in the preprocessing method. Instead of using the PTT, they introduced a regression model which uses the rPPG waveform feature indices and facial skin temperature as inputs for estimating BP (see Figure 15 for reference as well). This method acquired an MAE for MBP in a range of $\left[1.72,4.09\right]$ mmHg.

The authors of [218] estimated BP by extracting the PPG waveform through principal component analysis (PCA) [236]. They used PCA to calculate the component that carries signal intensity variation of the red channel in the video frames. They defined the PPG signal as the difference between the component of PCA and the raw video data in the red channel. The basic regression model with 20 features in the time and frequency domains as inputs was proposed to estimate BP. The AE was 3.72±5.08 mmHg for DBP and 3.90±5.37 mmHg for SBP. The authors of [225] stated that, in accord with Hooke’s law for calculating the fluid pressure of a wall, pressure waves are related to the peaks and valleys of the acquired signals. The body mass index (BMI) was used as a correction coefficient. They used one-channel processing on the green channel and the blind signal separation procedure of the JADE algorithm. The JADE algorithm separates the components which are the random estimates of the source signals in random order. Finally, they proposed a regression model based on the peaks and valleys of the signal processed by the JADE algorithm. The RMSE was 4.1665 for SBP and 2.8531 for DBP. Another study [224] derived a mathematical model to calculate the effects of hemoglobin, melanin, and light shadowing in sequential pictures. They removed the melanin and lighting effects from the video to extract the hemoglobin signal as an rPPG waveform by using knowledge of the melanin light absorption spectrum and camera spectral sensitivity. The waveform shape and PTT were collected from the rPPG signal to be the features to estimate BP. BP was estimated under two conditions: with and without body movement. The errors for SBP with and without body movement were −0.1±12.2 mmHg and −1.0±5.6 mmHg, respectively, while the errors for DBP with and without body movement were not estimated.

### 4.3. Deep Learning (DL) Methods

There are papers that propose models to predict BP, and most of them are based on waveform methods. Some of them use both waveform and PTT methods. A deep learning model can select features, and hence, most papers input several features, and the models take part in weighting the features. One study [229] performed ICA as the preprocessing method for rPPG and subsequently extracted the features from rPPG waveforms, including the systolic amplitude, pulse interval, systolic slope, diastolic slope, peak interval, crest time, and delta time ($a_{3}-a_{2}$). The authors proposed a feedforward neural network model which had one hidden layer for BP estimation. In the training phase, the extracted features were used as inputs. The experiment was set up with 2 phases—afternoon and evening—and was conducted with 20 human subjects. The error rate for SBP was 9.62% in the afternoon phase and 8.4% in the evening phase, and the error rate for DBP was 11.63% in the afternoon phase and 11.18% in the evening phase.

One study [230] selected the ROI by performing transdermal optical imaging (TOI). Since the color value can be in a range from 0 to 255, each channel from the three color channels contained 8 bitplanes which were 0 or 1. TOI is the technique for training a model to select bitplanes corresponding to hemodynamic changes. To calculate BP, 17 different facial regions were selected with the TOI method. The experiment was set up with over 1300 subjects, and 126 features related to transdermal blood flow were extracted from 17 different ROIs in the videos, such as the pulse amplitude, heart rate, pulse transit time, and pulse shape. As for the additional “meta-features”, 29 features were extracted to account for different imaging conditions and handle ambient room temperature and the demographic characteristics of the subjects. Then, 30 eigenvectors from conducting PCA to reduce the data dimensions were put into a multi-layer perceptron to calculate the BP. The errors for SBP and DBP were 0.39±7.30 mmHg and −0.20±6.00 mmHg, respectively.

BP is related to some physical characteristics, and some papers endeavored to extract the physical characteristics as the input features. To illustrate this, the authors of [230] personalized the model by inputting the meta-features, and the authors of [225] personalized the model by inputting the BMI. One study [147] compared models with and without personalization. The models were AlexNet, ResNet, and LSTM trained on PPG data. The models were fine-tuned with rPPG data and a leave-two-out cross-validation scheme. They conducted three training scenarios. The first scenario, which fine-tuned the model without personalization, was to use 15 out of 17 subjects for fine-tuning the model, while the remaining two were conducted for validation and testing. Secondly, they fine-tuned the model with personalization by using the first 20% of the measurement samples of the subjects for testing. In the last training scenario, 20% of the samples from the whole measurement in testing were drawn randomly for fine-tuning. Consequently, fine-tuning models by using rPPG yielded a significant improvement in the MAE, especially regarding SBP but also DBP. Personalizing the training caused a moderate improvement in the MAE for some model architectures. For AlexNet, the MAEs of SBP and DBP were 14.2 mmHg and 10.7 mmHg, respectively. For ResNet, the MAE for SBP was 12.7 mmHg, and that of DBP was 10.8 mmHg. For LSTM, the MAE for SBP was 14.4 mmHg, and the MAE for DBP was 10.5 mmHg. However, personalization makes the training costs expensive when the subject size is huge because the model has to be trained with all the subject samples.

To dominate the disadvantage of personalization, one study [153] proposed a model selector by mapping the BMIs and ages of subjects to their SBP (see Figure 16). A model selector would select the responding model from 10 trained models for each subject by using BMI and age. Hence, the authors trained 10 models instead of 1 model per subject. Since the lack of data is an obstacle to the training procedure and evaluation, they proposed synthetic data generation with InfoGAN by learning the information about the samples’ noise and features. The model is an encoder-decoder architecture whose features are the chrominance-based (CHROM) [231] rPPG signals of the upper and lower face. The obtained AE was 9.13±8.18 mmHg for SBP, and the AE was 8.76±6.13 mmHg for DBP. Another study [232] calculated the continuous BP by frequently tracking the spatial information of facial pulse waves (see Figure 17). Transformers from the extracted hemoglobin videos to spatial information, pulse contour descriptors, and spatial pulse contour descriptors were created. The authors proposed a CNN for estimating BP from the spatial information of facial pulse waves. They obtained an MAE for SBP of 6.7 mmHg and an MAE for DBP of 5.4 mmHg.

## 5. Discussion

Based on the above investigations, several innovations have been forthcoming in the area of BP measurement. On the analytical side, it initially started from modeling with an electric circuit and modeling by Navier–Stokes and continuity equations and progressed to modeling by the same equations with the material properties of blood vessels and the topology of the vessel system. On the data-driven side, it commenced with the PTT by simple regression due to a biological relationship between the PWV and BP, through DNN models of PPG, and to the same ML development on rPPG.

In the measurement from PPG signals, a CNN-LSTM model appeared to achieve acceptable performance, but it requires complex preprocessing to extract the PAT calculated from ECG and PPG waveforms as one of the inputs in the training phase of the model, because the PAT contains patient-specific body characteristics and pathological conditions. If we only consider the PPG waveforms and BP as the available data source, a U-Net model can achieve sound performance. However, the number of subjects in both models is relatively small compared with other models, which were verified by a larger number of subjects and yielded just a little bit worse accuracy. On the other hand, in the measurement from rPPG signals, regressions in traditional machine learning with adequate preprocessing can give satisfactory results. Although the features of rPPG waveforms are only used to train the models, the number of subjects for the experiments is relatively small for verification. In the deep learning attempt, a CNN model trained by the spatial patterns of facial pulse waves appears to be good, but the number of testing subjects is still very small. Hence, if one really wants to adopt those models, one should make sure of the robustness of the datasets and verify their feasibility if necessary.

## 6. Future Directions

Based on the previous work, we can propose that BP measurement via rPPG will be the common trend. We suggest some useful directions, which are as follows.

### 6.1. Satisfactory Signal Quality

The quality of rPPG should be improved, because better quality rPPG signals will ensure a more accurate BP measurement [201]. A suitable optimized algorithm should be found to transfer rPPG to PPG, because most of the features used for model establishment originate from PPG and are selected based on the physiological motivation. The authors of [237,238,239] noted that to extract stronger rPPG signals from the pixels of videos, the areas near the cheeks and forehead should be chosen. After the normalization of the pixel values, one of the useful denoising techniques is to extract all the frequency bands (including all their corresponding harmonic frequencies) related to the human body, as these can be relied upon to locate vital signs. A research issue that can be posited is how large the chosen bandwidths should be so as to obtain sharper rPPG signals comparable to the corresponding PPG signals. Since the models using PPG signals to measure BP were better studied than those models using rPPG signals to measure BP, it will be beneficial to the development of the research field if it is possible to change noisy rPPG signals into PPG signals.

### 6.2. Public Dataset Enlargement

Public standardized datasets should be enlarged. Here, “standardized” can be defined as collecting as much information on patients as possible, including but not limited to height, weight, age, PPG waveforms, and habits. To date, there have been five available public datasets, as shown in Table 5. First, the MIMIC dataset [124], created under the auspices of the National Center for Research Resources of the National Institutes of Health, includes data recorded from over 90 ICU patients and 121 records, and it has enough information for studying BP, including ECG, PPG, and heart rate. Second, the MIMIC-III dataset [240] collected the clinical data of patients who were admitted to a medical center in Boston, Massachusetts. It incorporates the data contained in MIMIC-II and augments MIMIC-II with 621 newly collected data between 2008 and 2012. The dataset covers 38,597 adult patients and 49,785 hospital admissions. For statistical information, the median age of the patients was 65.8 years ($Q_{1}–Q_{3}:52.8–77.8$), and 55.9% of the patients are men. The number of samples per subject spans a broad range from just a few hundred to over 500,000 [147]. Moreover, another study [147] provided a subset of the MIMIC-III dataset called MIMIC-B, which denotes the number of samples for every subject. MIMIC-B has a total pool of 4000 records and approximately 150 million samples. The authors of [35,126] provided a preprocessed MIMIC-II dataset. They smoothed the PPG and ECG signals and discarded the unacceptable signals. It is freely available on both Kaggle (the hyperlink to the Kaggle dataset is https://www.kaggle.com/datasets/mkachuee/BloodPressureDataset?resource=download (accessed on 15 September 2022)) and from the UCI Machine Learning Repository [144]. The third dataset is the UQVS dataset (the hyperlink to the UQVS dataset is https://outbox.eait.uq.edu.au/uqdliu3/uqvitalsignsdataset/index.html (accessed on 15 September 2022).) [113], which contains 32 subjects (25 general anesthetics, 3 spinal anesthetics, and 4 sedations). Its monitoring data is in a range from 13 min to 5 h (the median duration is 1 h and 45 min). The next dataset, the Figshare dataset [139], contains 219 subjects and a total of 657 PPG waveform segments. It provides detailed individual information including the age, sex, height, weight, heart rate, and BMI for each subject. However, the original MIMIC dataset [138] does not provide that general information. Lastly, the authors of [122] provided HR, ${\mathrm{SpO}}_{2}$, and BP data from 22 subjects (13 females and 9 males) aged between 18 and 78 years old, weighing between 50 and 94 kg, and with heights from 160 to 195 cm.

However, None of these datasets have videos of human faces for the extraction of rPPG signals. Even though there are two public datasets, the HCI-tagging database and MAHNOB database (This dataset contains videos of 30 subjects with 6 video cameras, a head-worn microphone, an eye gaze tracker, as well as physiological sensors measuring ECG, EEG (32 channels), the respiration amplitude, and skin temperature. For details, browse the website https://mahnob-db.eu/hci-tagging/ (accessed on 15 September 2022). We did not include this dataset in Table 5 either, since it does not have BP data.) [242] and UBFC-RPPG Dataset (This dataset provides videos of human faces for rPPG signals, PPG signals, HR and ${\mathrm{SpO}}_{2}$. For details, browse the website https://sites.google.com/view/ybenezeth/ubfcrppg (accessed on 15 September 2022). We do not put it into Table 5 since it does not include BP data.) [243], for the verification of models between rPPG and PPG, they do not have BP data. Thus, in order to easily verify models from rPPG to BP and further studies among rPPG, PPG, and BP, a dataset with videos of human faces, PPG signals, and BP data with plenty of subjects should be prepared. In particular, personal data including but not limited to race, gender, age, and medical history should also be collected, since PPG waveforms of patients in various categories may have some differences. (For differences in PPG pulses, see [244].) Furthermore, it is highly recommended that videos of faces, PPG signals, and the vital signs of each subject during each follow-up medication check be sustainably recorded because they are useful for personal calibration in BP measurement. Of course, the privacy problem is the first paramount issue that researchers need to solve before obtaining such valuable and specific data.

### 6.3. Effective Calibration by Personalization

Apart from the above, an effective model with calibration and personalization should be developed. Recent works have started to study this. For PTT calibration, the authors of [245] performed a calibration experiment with a 24-h interval. Their results show that models involving the photoplethysmogram intensity ratio (PIR) have less of a difference in error between calibrations. The authors of [51] mentioned the necessity to calibrate the phase shifts on different PPG sensors for PTT calculation. As for other calibration examples, the authors of [246] introduced five equations for calibrating SBP and DBP measured by a random zero sphygmomanometer (RZS) BP measurement device and an oscillometric device (OD), respectively, in the Jackson Heart Study (JHS): ignoring the change, ordinary least squares (OLS) regression, adding the average difference, Deming regression, and robust regression. It was demonstrated that robust regression gives a higher $R^{2}$ statistic in the JHS. The authors of [59] introduced the calibration methods of BP devices. The authors of [247] reviewed the linear approximation of BP calibration. It was mentioned that the short-term validity of linear PTT models and the calibration need to be updated frequently under various conditions. The authors of [248] proposed a nonlinear ML-GPR model to estimate the regression between BP values and PPG features by grouping the age range of a user. They pointed out that calibrations can be implemented in an embedded system for personalized measurement and be adapted to different environments and health statuses. Thus, from the above, most of the calibration focuses on the fine-tuning of coefficients in regression equations, which depends on the datasets.

Another type of calibration that has captured a lot of attention is called personalization, which means fine-tuning the models by subject-specific data. Some works have been carried out for personalization. For example, the authors of [249] proposed a transfer learning technique (called BP-CRNN-Transfer) that personalizes specific layers of a network pretrained with abundant data from other patients. The MAE of SBP was 3.52 mmHg, and that of DBP was 2.20 mmHg. The authors of [147] used 20 percent of the non-mixed test set to fine-tune the pretrained neural architecture of the main model and validated the model with the remaining 80% of the test set. This highlightd that persionalization significantly reduced the prediction errors. The authors of [133] also conducted the same experiment on PPG and ECG previously. The authors of [250] investigated the effect of selective single-parameter personalization on the performance of multi-parameter models for pulse arrival time (PAT)-based blood and found that parameter personalization is key to enhancing tracking performance. Most of them require high computing power, which is costly if the calculation neural network is put in the cloud or on a smartphone, and privacy is another issue for personalization. The authors of [251] raised an idea called federated learning (FL) to solve the privacy issue and achieve personalization, which means exchanging the weights applied on the training model across multivarious users via a central server, while users’ data are stored on their own local devices without sharing. Thus, a privacy-protected model with correct inputs and parameters and personalized calibration is more applicable to the reality of the ML industry.

## 7. Conclusions

To summarize, this review discussed the historical development of BP measurement, starting from biophysical theory, through contact-based BP measurement from PPG signals, and to contactless BP measurement from rPPG signals. Since the ultimate goal of this research field is to predict BP accurately based on rPPG signals, using the above discussions, we offer a range of proposals in model training from rPPG signals in the future. Initially, we suggest that we can still train neural networks with adequate preprocessing of rPPG signals to predict BP, since many DNN models have not yet been examined. A further recommendation is that we adopt personal information to assist with model training. From the experience of using the PAT, we note that the PAT originated from specific personal body characteristics which are strongly correlated with personal information, including but not limited to, age, BMI, and habits. This information allows us to gain increased accuracy in readings. However, the verification and accuracy of a large dataset are very important. Although contactless BP measurement development is still emerging, it will continue to be a contentious area of research, especially in the post-pandemic era.

## Figures and Tables

**Figure 1 healthcare-10-02113-f001:**
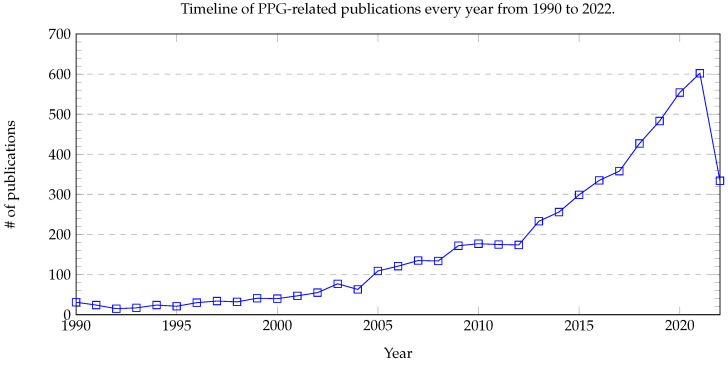
A timeline of PPG-related publications every year from 1990 to 2022. For 2022, the number of publications was recorded up to 26 July 2022.

**Figure 2 healthcare-10-02113-f002:**
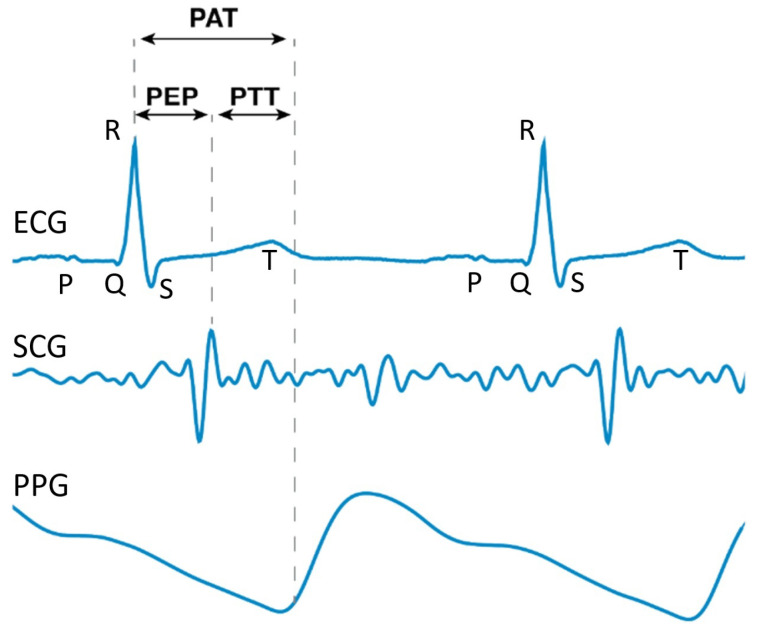
This figure shows waveforms from various measurements and the relation between PAT and PTT as in Equation (Equation 1). ECG = electrocardiography, SCG = seismocardiography, and PPG = photoplethysmography. The letters in the ECG waveforms represent the parts of the waveforms.

**Figure 3 healthcare-10-02113-f003:**
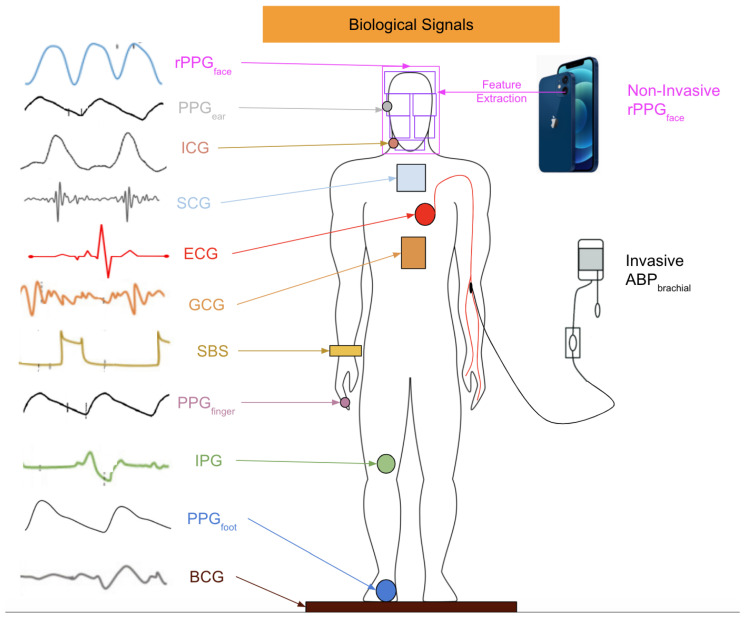
Biological signals from various parts of the body. This figure was modified and enhanced from [52]. The abbreviations represent BP measurement signals at various parts on the human body. BCG = ballistocardiography, PPG = photoplethysmography, IPG = impedance photoplethysmography, SBS = strain-based sensor, GCG = gyrocardiography, ECG = electrocardiography, SCG = seismocardiography, ICG = impedance cardiography, ABP = arterial blood pressure (by invasive measurement on the arms), and rPPG = remote photoplethysmography (by cameras).

**Figure 4 healthcare-10-02113-f004:**
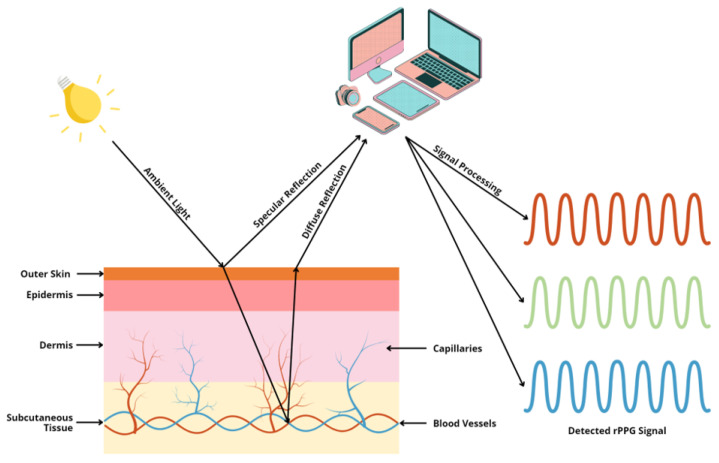
A working principle of rPPG. This figure was modified from [53]. The specular component gives the information on the skin’s surface, which does not have any physiological signals. The diffused counterpart gives the subtle change in blood flow, which provides physiological information after meticulous signal processing.

**Figure 5 healthcare-10-02113-f005:**
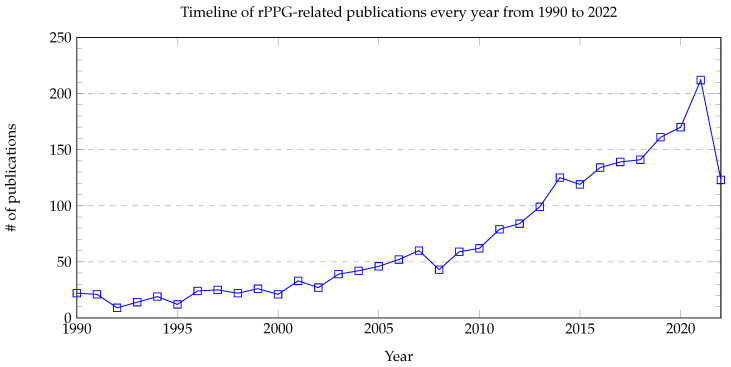
A timeline of rPPG-related publications every year from 1990 to 2022. For 2022, the number of publications was recorded up to 26 July 2022.

**Figure 6 healthcare-10-02113-f006:**
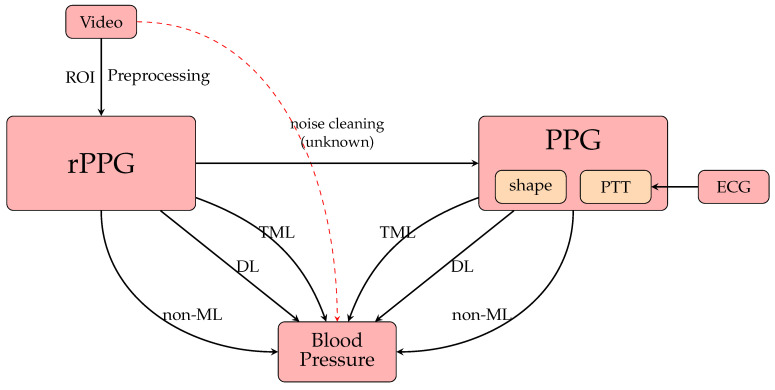
A flowchart showing the research situation in BP measurement. There are 5, 11 and 30 pieces of work using non-ML, ML and DL methods respectively to predict BP from PPG signals, while there are 4, 8 and 7 pieces of work using non-ML, ML and DL methods respectively to predict BP from rPPG signals. ROI preprocessing means preprocessing of data received from some regions of interest on a human face recorded on video. The red dotted line means that there are no research papers talking about the techniques from a video for BP measurement.

**Figure 7 healthcare-10-02113-f007:**
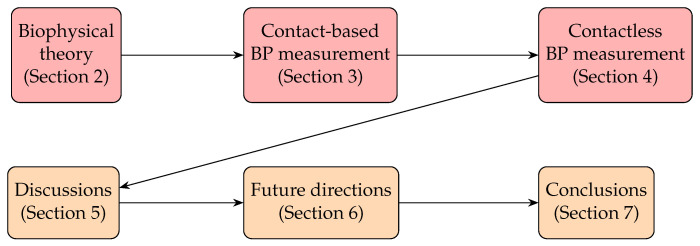
A flowchart showing the organization of this review paper.

**Figure 8 healthcare-10-02113-f008:**
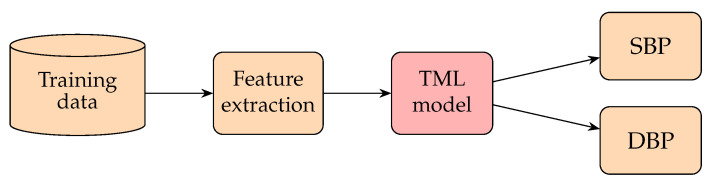
A pipeline of traditional machine learning (TML) methods for PPG signals. Features inside training data are extracted based on various author-preferred criteria (e.g., highest correlation with the ground truth BP) to train their studied TML models and give the predictions for SBP and DBP.

**Figure 9 healthcare-10-02113-f009:**
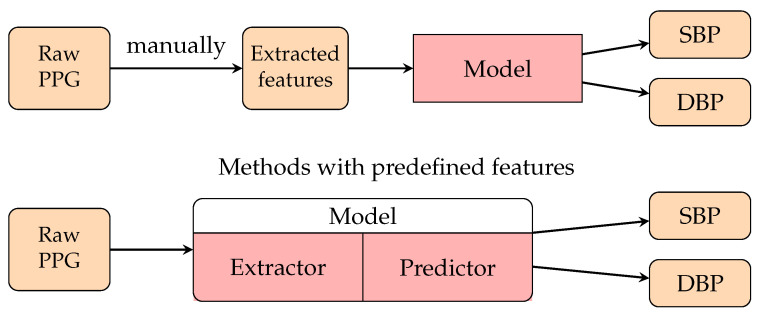
The flowchart of general PPG waveform-based methods. (upper flowchart) [123,125,127,128,136]. Features inside training data of raw PPG signals are manually extracted to train their studied models and give the predictions for SBP and DBP. (lower flowchart) [133,135,137,145]. The whole waveform profiles of raw PPG signals are put into models, which automatically pick up useful features and give predictions for SBP and DBP.

**Figure 10 healthcare-10-02113-f010:**
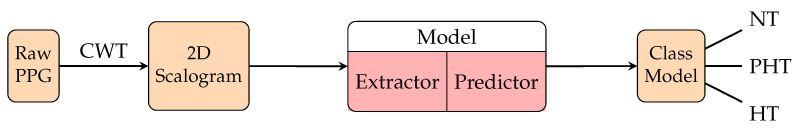
The structure of the algorithm in [131]. PPG signals are preprocessed by continuous wavelet transform (CWT) to produce two dimensional scalograms for identifying the low-frequency or fast-changing frequency components. These scalograms are converted into RGB images for the convolutional neural network pretrained by GoogLeNet. This model automatically extracts features in the scalograms to predict the BP categories: normotension (NT), prehypertension (PHT), and hypertension (HT).

**Figure 11 healthcare-10-02113-f011:**
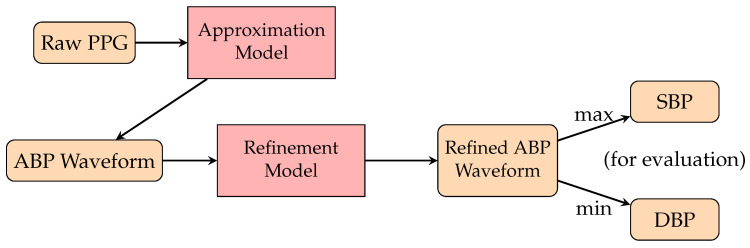
The structure of the algorithm in [141]. PPG signals are preprocessed by the wavelet transform with the removal of the very low and very high frequency components, the soft Rigrsure thresholding [198,199], and the normalization. The filtered PPG signals are processed through the approximation network (one-dimensional deep supervised U-net model), which estimates the ABP based on the inputs, and then refined by the refinement network. The maximum and minimum of the refined ABP signals are taken as SBP and DBP, respectively.

**Figure 12 healthcare-10-02113-f012:**
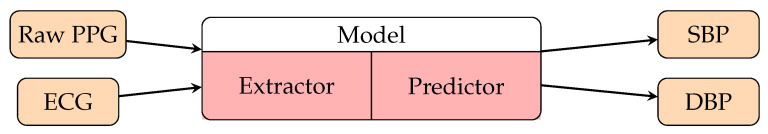
The flowchart of general PTT-based DL methods adopted in [132,140,142,143,146]. After preprocessing, PPG and ECG signals are fed into the models, which automatically extract the relevant features from the input signals and predict SBP and DBP.

**Figure 13 healthcare-10-02113-f013:**
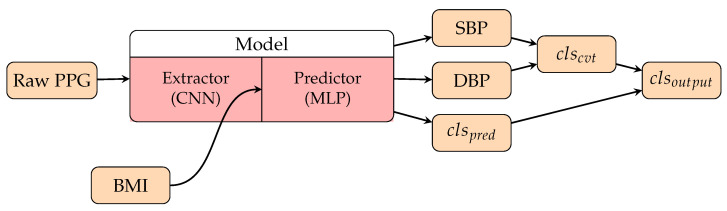
The structure of the algorithm in [138]. The PPG signals are directly put into the CNN part of the model for feature extraction. The model uses the MLP part with the aid of the BMI of the corresponding subjects to predict SBP, DBP, and the hypertension classes. Here, $cls_{pred}$ is the predicted class from the model, $cls_{cvt}$ is the converted class from the regression result, and $cls_{output}$ is the final output class of the algorithm.

**Figure 15 healthcare-10-02113-f015:**
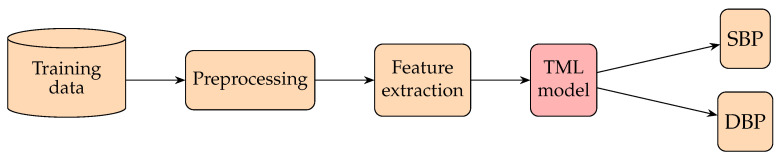
A pipeline of traditional machine learning (TML) methods for rPPG signals adopted in [220,223]. Those rPPG training data are initially preprocessed. Those relevant features of the filtered signals are extracted manually to feed their target models to predict SBP and DBP.

**Figure 16 healthcare-10-02113-f016:**
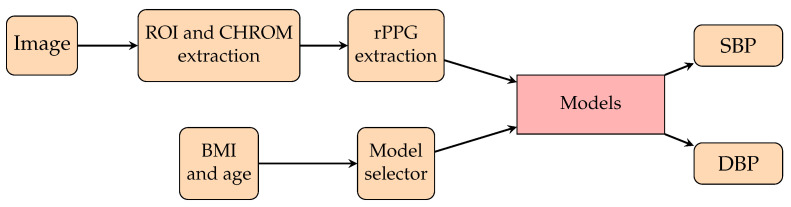
A pipeline of a deep learning model for rPPG signals proposed in [153]. On one hand, pixels from the ROIs of face images are extracted by CHROM as the rPPG signals. On the other hand, age and BMI are fed into another model selector to promote the model training to predict SBP and DBP.

**Figure 17 healthcare-10-02113-f017:**
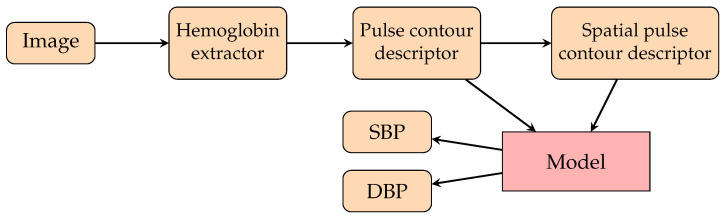
A pipeline of a deep learning model for rPPG signals proposed in [232]. Pixels from RGB face images are extracted to form intensity maps of melanin, hemoglobin (Hem), and shading (residual information). Those extracted signals are used for constructing the pulse contour descriptor, which does not include spatial information, and the spatial pulse contour descriptor, which includes spatial details, in order to preserve spatial phase relationship of the rPPG signals. Both descriptors are used as inputs for the model for prediction of SBP and DBP.

**Table 1 healthcare-10-02113-t001:** Different stages of cardiovascular (CV) health. For details on treatment, see [1,3,4,5,6,7,8].

CV Stages	SBP (mmHg)	DBP (mmHg)	Additional Biological Information
Hypotension	<90	<60	Low BP leading to oxygen deprivation in organs, resulting in tissue necrosis. May induce shock and cardiac arrest [9]. Symptoms: dizziness, tiredness, back pain, heart palpitations, etc. Usually a common side effect of drug therapies (e.g., beta blockers and diuretics) [10].
Normotension (NT)	90–120	60–80	BP may fluctuate based on poor lifestyle habits. Examples: lack of exercise, fatty diet, anxiety, insomnia, alcoholism, aging, etc. [11].
Prehypertension (PHT)	121–139	81–89	BP is higher than normal but not within range of stage 1 hypertension, also known as high-normal BP. Known as the upper range of healthy BP that determines the future risk of clinically overt hypertension [3], it is further divided into its own 1st and 2nd stages to further define hypertensive risk parameters [12].
Hypertension I (HT-I)	140–159	90–99	BP is high enough to be a risk factor. Occurs when the heart is overly stressed. Treatment may not be required, but drug therapy will significantly reduce BP. SBP/DBP range refers to daytime BP as sleep naturally lowers SBP and DBP [13].
Hypertension II (HT-II)	160–179	100–109	BP is very high, and CVD is very likely. Lack of treatment may likely result in end organ failure and permanent damage. Commonly found in elderly people. SBP control primarily determines risk of CVD and death [4].
Hypertensive Crisis or Urgency	>180	>110	BP is fatally high, and premature death is likely [5]. Symptoms include chest pain, numbness, weakness in limbs, blurred vision, breathing difficulty, and other symptoms associated with stroke or myocardial infarction [14]. Immediate treatment in ICU is recommended for rapid reduction in BP. Acute cardiac, renal, and neural damage may occur if treatment is too late [6].

**Table 3 healthcare-10-02113-t003:** BHS standard. The percentage represents the cumulative frequency of error.

Cumulative Frequency of Error	≤5 mmHg	≤10 mmHg	≤15 mmHg
Grade A	60%	85%	95%
Grade B	50%	75%	90%
Grade C	40%	65%	85%

**Table 5 healthcare-10-02113-t005:** Datasets used for verification of models in BP.

Dataset	Ref.	Year	Number of Subjects	Number of Subjects (Male)	Number of Subjects (Female)	Age	Race	Remarks
MIMIC I	[124]	2000	90	-	-	-	-	ICU patients
MIMIC III	[240]	2016	53,423	27,983 (55.9%)	25,440 (44.1%)	median =65.8 $\left(Q_{1}–Q_{3}:52.8–77.8\right)$	-	Includes MIMIC II
Figshare dataset	[139]	2018	219	48%	52%	21–86, 61 subjects $\in \left[50–59\right]$	-	23.7%= hypertension
UQVS	[113]	2012	32	-	-	-	-	-
Dataset without a name	[241]	2019	22	9	13	18–78	-	Weight: 50–94 kg, height: 160–195 cm

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
