# Peer review of "Blood Pressure Measurement: From Cuff-Based to Contactless Monitoring"

_healthcare, 2022, doi:10.3390/healthcare10102113_

Round 1
Reviewer 1 Report
This paper gives a comprehensive review of the blood pressure measurement from cuff-based to contactless monitoring. The directions raised in Section 5 are valuable. The best algorithms are given under different conditions and future directions are pointed. The manuscript is well structured and well written. The following issues should be addressed.
1. Similar papers reviewing BP measurements should be included and discussed in Introduction.
2. Page 30, it seems that the five datasets provide adequate videos, number of subjects, personal information, etc., whereas in the previous paragraph, the authors said that public standardized datasets should be enlarged. The connection between these two paragraphs should be enhanced.
3. Page 31, how do the authors know in the measurement from PPG signals, a CNN-LSTM model can achieve the best performance. If the authors determined based on the results reported in the original papers, the results may not be reliable because the experiments are not conducted on the same settings.
Author Response
Dear reviewer,
Thank you very much for your kind reminder. We tackle your requirements as the following.
- Similar papers reviewing BP measurements should be included and discussed in Introduction.
Response: We added the relevant review papers and discussed them briefly in the introduction.
- Page 30, it seems that the five datasets provide adequate videos, number of subjects, personal information, etc., whereas in the previous paragraph, the authors said that public standardized datasets should be enlarged. The connection between these two paragraphs should be enhanced.
Response: We updated Section 6.2 public dataset enlargement to strengthen the connection between the present public datasets and the missing parts that are important for further studies of BP measurement.
- Page 31, how do the authors know in the measurement from PPG signals, a CNN-LSTM model can achieve the best performance. If the authors determined based on the results reported in the original papers, the results may not be reliable because the experiments are not conducted on the same settings.
Response: Yes, we agree with you. Since our judgments are based on what the original papers reported, we change the wordings into "appears to", "seems to". But, we also emphasize that the number of subjects may not be enough to remind readers further check the models in their own verification datasets if they really want to adopt their results. Also, in order to satisfy the requirements of reviewer 2, we adjust the position of this paragraph to section "Discussion".
We sincerely hope that you can review and reconsider our revised version. Please do not hesitate to give us more questions and/or comments so that we can respond to them promptly.
Best regards,
MAN Ping Kwan
Reviewer 2 Report
Blood pressure (BP) determines whether a person has hypertension and offers implications as to whether he suffers from cardiovascular disease. Cuff-based sphygmomanometers have traditionally provided both accuracy and reliability, but they require bulky equipment and relevant skills to obtain precise measurements. BP measurement from photoplethysmography (PPG) signals has become a promising alternative for convenient and unobtrusive BP monitoring. Moreover, the recent developments in remote PPG (rPPG) algorithms have enabled new innovations for contactless BP measurement.
The authors illustrateed the evolution of BP measurement techniques from the biophysical theory, through the development of contact-based BP measurement from PPG signals, to the modern innovations of contactless BP measurement from rPPG signals.
The authors stated to have consolidated knowledge from a diverse background of academic research to highlight the importance of multi-feature analysis for improving measurement accuracy.
They concluded with the current challenges, opportunities and future directions in this emerging field of research.
The paper is interesting and written with enthusiasm.
However in the present form it is very hard to read/follow.
Please rewrite according to the standards: Introduction, M&M, results, discussion, cosnclusion.
Further suggestions are the following:
1. Insert a table with the acronyms.
2. Insert a clear purpose.
3. Insert the methods describing the design of the study and a flow chart allowing the reader to follow the rationale.
4. Sections 2-4 are results of the study. Insert into a new section results after the methods.
5. Each figure (for example the models) must be described in details.
6. Insert the limitations in the discussion.
7. Check the mdpi standard for the text and the tables.
8. Avoid short paragraphs (as for example par 5.1)
I’ll be happy to read the revised and improved manuscript
Author Response
Dear reviewer,
Thank you very much for your kind reminder. We tackle your requirements as the following.
- Please rewrite according to the standards: Introduction, M&M, results, discussion, and conclusion.
Response: Since this paper is a review paper, we follow the requirements of the layout suggested by MDPI as shown in section 2.1.15 in “https://www.mdpi.com/authors/layout#_bookmark71”.
- Insert a table with the acronyms.
Response: We added an acronym table in the section “Abbreviations”.
- Insert a clear purpose.
Response: We added the purposes at the end of section “Introduction” by writing “Because of the fact that there are a multitude of productive research progresses in BP measurement, it is useful if there are some reviews summarizing key techniques and giving the prospective future directions to motivate the development. Particularly, now that contactless BP measurement has been emerging recently, it is a good time to summarise what types of contactless BP measurement has been done and to give effective implications to the industry to overcome the challenges.”
- Insert the methods describing the design of the study and a flow chart allowing the reader to follow the rationale.
Response: Same as the response of question 1. We also inserted the flowchart of this review paper in figure 7 to show the structure of this paper.
- Sections 2-4 are results of the study. Insert into a new section results after the methods.
Response: Same as the response of question 1.
- Each figure (for example the models) must be described in details.
Response: We supplemented more explanations of the structures of the flowchart algorithms in details.
- Insert the limitations in the discussion.
Response: For the limitations of models, we ever thought about how to write about this part. We think that it is difficult to fairly compare among models because different papers use different feature extractions, training procedures, the number of layers, model parameters, verification datasets and the number of subjects. Thus, we cannot say anything about the limitations of models. If you have some ideas, please feel free to let us know.
- Check the mdpi standard for the text and the tables.
Response: We followed the suggested section names and layout(tables) in mdpi review requirement.
- Avoid short paragraphs (as for example par 5.1)
Response: We enlarged some short paragraphs like the paragraph in section 5.1. However, we kept some short paragraphs due to different catergorisations and easier reading. If you have different views towards some specific paragraphs, please feel free to let us know.
We sincerely hope that you can review and reconsider our revised version. Please do not hesitate to give us more questions and/or comments so that we can respond to them promptly.
Best regards,
MAN Ping Kwan
Round 2
Reviewer 2 Report
The ms improved